# BAG3 Positivity as Prognostic Marker in Head and Neck Squamous Cell Carcinoma

**DOI:** 10.3390/cancers17111843

**Published:** 2025-05-31

**Authors:** Pietro De Luca, Francesco Antonio Salzano, Angelo Camaioni, Leopoldo Costarelli, Raul Pellini, Gerardo Petruzzi, Renato Covello, Luigi Vittori, Filippo Ricciardiello, Giuseppe Ricciardiello, Alessandro Iacobelli, Anna Lisa Cammarota, Paola Manzo, Jelena Dimitrov, Arianna Mauro, Margot De Marco, Liberato Marzullo, Alessandra Rosati

**Affiliations:** 1Otolaryngology Department, Isola Tiberina-Gemelli Isola Hospital, 00186 Rome, Italy; pietro.deluca.fw@fbf-isola.it; 2Department of Medicine, Surgery and Dentistry “Schola Medica Salernitana”, University of Salerno, 84081 Baroniss, Italy; frsalzano@unisa.it (F.A.S.); acammarota@unisa.it (A.L.C.); pamanzo@unisa.it (P.M.); jdimitrov@unisa.it (J.D.); marzullo@unisa.it (L.M.); 3Head and Neck Department, Nostra Signora della Mercede Clinic, 00198 Rome, Italy; camaioniangelo@libero.it; 4Pathology Unit, San Giovanni-Addolorata Hospital, 00184 Rome, Italy; lcostarelli@hsangiovanni.roma.it; 5Department of Otolaryngology-Head and Neck Surgery, IRCCS Regina Elena National Cancer Institute, Istituti Fisioterapici Ospitalieri (IFO), 00144 Rome, Italy; raul.pellini@ifo.it; 6Department of Pathology, IRCCS Regina Elena National Cancer Institute, 00144 Rome, Italy; petruzzi.gerardo@ifo.it (G.P.); covello@ifo.it (R.C.); 7Department of Radiological, Oncological and Pathological Sciences, Sapienza, University of Rome, 00185 Rome, Italy; luigi.vittori@uniroma1.it; 8Otolaryngology Department, AORN Cardarelli, 80131 Naples, Italy; filippo.ricciardiello@aocardarelli.it (F.R.); ricciarello.gius@libero.it (G.R.); 9Patology Unit, AORN Cardarelli, 80131 Naples, Italy; alessandro.iacobelli@aocardarelli.it; 10Cytokine Lab, Clinical Pathology- Hospital Unit “Gaetano Fucito”, University Hospital “San Giovanni di Dio e Ruggi d’Aragona”, 84085 Mercato San Severino, Italy; arianna.mauro27@gmail.com; 11FIBROSYS s.r.l., Academic Spin-Off, University of Salerno, 84081 Baronissi, Italy

**Keywords:** BAG3, head and neck squamous cell carcinoma, prognostic marker, immunohistochemistry

## Abstract

Head and neck cancers exhibit considerable heterogeneity, complicating the prediction of disease progression and treatment response. Researchers are actively investigating reliable biomarkers to forecast disease trajectories and inform therapeutic decisions. This study examines the role of BAG3, a protein involved in cell survival and stress response, as a potential prognostic marker in head and neck cancers. By analyzing BAG3 expression across various head and neck cancer types and correlating it with disease-free survival, the study aims to elucidate BAG3 positivity’s influence on cancer progression. Our findings indicate that immunohistochemical testing for BAG3 positivity could aid in diagnosing tumor progression and prognosis in head and neck squamous cell carcinoma (HNSCC), with potential implications for therapeutic strategies.

## 1. Introduction

Head and neck cancers (HNCs) encompass a wide array of malignancies that develop in various anatomical sites, including the oral cavity, pharynx, larynx, nasal cavity, and salivary glands [1,2]. The predominant type of these cancers is squamous cell carcinoma (HNSCC), which is frequently linked to risk factors such as tobacco and alcohol consumption, as well as human papillomavirus (HPV) infection [3,4]. Although there have been significant advancements in multimodal treatment strategies—such as surgery, radiation therapy, chemotherapy, and targeted therapies—patient outcomes continue to vary considerably due to the diverse nature of these tumors [5,6]. Histopathological biomarkers are crucial in head and neck cancers for improving prediction and prognosis. They offer detailed insights into tumor behavior, aiding personalized treatment decisions, risk stratification, and monitoring, thereby enhancing patient outcomes. These biomarkers, discovered through methods like immunohistochemistry and molecular profiling, offer valuable insights into tumor biology, aggressiveness, and responses to treatment [7,8]. The BAG3 (Bcl-2-associated athanogene 3) protein is a multifunctional co-chaperone of Hsp70 protein with an important involvement in tumor biology, particularly in cancer cell survival, resistance to therapy, and tumor development [9]. Its expression is normally raised in normal cells during stress, but it is constitutively overexpressed in a variety of human malignancies, including glioblastomas [10], pancreatic adenocarcinomas [11], melanomas [12], breast cancers [13], and thyroid carcinomas [14]. BAG3 contributes to tumor progression through several mechanisms, including anti-apoptotic activity, chemoresistance, promotion of tumor invasiveness, and maintenance of cancer stem cells [15]. In glioblastoma cells, BAG3 interacts with heat shock protein 70 (HSP70) to limit apoptosis by preventing the pro-apoptotic protein BAX from translocating to mitochondria [10]. Similarly, in thyroid carcinomas [14], leukemia [16], and ovarian cancer cells [17], BAG3 downregulation makes cells more susceptible to apoptosis-inducing therapy. Indeed, scientific evidence increasingly supports the link between high BAG3 expression and chemotherapy resistance across various malignancies. In ovarian cancer, for example, elevated BAG3 levels are associated with poor prognosis and resistance to treatments like paclitaxel and cisplatin [18], while in triple-negative breast cancer (TNBC), BAG3 overexpression contributes to apoptosis resistance and enhances cytoprotective autophagy, making these cells more resilient to chemotherapeutic agents [19]. Melanoma studies have also revealed that BAG3 positivity correlates with tumor aggressiveness and sustains NF-κB activation, further contributing to chemotherapy resistance [20]. Furthermore, BAG3 expression levels frequently correspond with tumor grade and aggressiveness. As an example, BAG3 expression in glioblastomas increases with tumor grade, and BAG3 silencing by siRNA inhibits cell growth both in vitro and in vivo [10], while elevated BAG3 levels in pancreatic ductal adenocarcinoma (PDAC) [11] and medulloblastoma [21] are associated with shorter patient survival. In PDAC, pancreatic cancer cell-secreted BAG3 activates macrophages via IFITM2-mediated signaling pathways such as PI3K and p38 MAPK phosphorilation, increasing tumor development and metastasis in a way that can be inhibited by a specific anti- BAG3, neutralizing monoclonal antibody [22]. Moreover, secreted BAG3 neutralization can enhance immunotherapy effects [23,24] while reducing fibrotic stroma [25] in PDAC animal models. Currently, there is limited specific information available on how BAG3 overexpression affects distinct subtypes of head and neck cancers. One risk factor for HNSCC is HPV positivity and patients with HPV-positive head and neck cancer (HNC) generally experience better survival outcomes; however, if recurrence occurs, it is more likely to happen in distant parts of the body and may arise later compared to HPV-negative patients [26]. A previous study showed that in HPV18-positive HeLa cells, BAG3 interacts with the viral E6 oncoprotein, stabilizing E6 and promoting degradation of the tumor suppressor p53. Downregulating BAG3 reduced E6 levels and restored p53, implicating BAG3 in HPV-driven cell transformation [27]. Furthermore, evidence suggests that BAG3 plays a key role in the tumor microenvironment, especially in fibrotic tumor phenotypes frequently seen in HNSCC [28,29,30]. Specifically, analysis using the GEPIA database revealed that BAG3 mRNA expression is higher in tumors compared to normal tissues across multiple cancers, including HNSCC, and that elevated BAG3 levels are associated with poorer clinical outcomes [28].

In this study, the expression of BAG3 has been investigated by immunohistochemistry in a series of head and neck cancers from different anatomical regions. The results were correlated with clinicopathological parameters of the tumors, including localization, tumor grade, tumor stage and survival, with the intention of providing additional tools to clinicians for predicting disease outcomes and guiding treatment decisions.

## 2. Materials and Methods

### 2.1. Patients

Patients with confirmed biopsy of SCC of oral cavity, oropharynx and larynx between January 2022 and January 2023 were retrospectively included in this study. The cohort included patients from multicenter sites, recruited in the departments of Otolaryngology—Head and Neck Surgery of San Giovanni Addolorata Hospital (Rome, Italy), IRCCS Regina Elena National Cancer Institute (Rome, Italy), and AORN Cardarelli (Naples, Italy). We conducted a retrospective review of patients’ medical records, considering the type of surgery performed, the histopathological results and the follow-up reports. For each patient, the following data were collected: gender; age at diagnosis; type of surgery performed (including any neck dissection performed at the time of the first surgery); presence of clinically evident lymph node(s) (cN) or distant metastasis at diagnosis; positive or negative tumor resection margins post-surgery; use of adjuvant radio chemotherapy (RTChT); presence of locoregional or distant recurrence; and health status at follow-up (including follow-up time in months). The TNM classification system (version 8) [31] was used to stage the tumor. Data on survival outcomes were extracted from mortality registries, outpatient visit notes and radiological follow-ups. The data collected from each center were analyzed. The exclusion criteria were as follows: (i) absence of follow-up, (ii) changes in the histological diagnosis during review of the slides, and (iii) slides unavailable for review. The data were inserted into a shared Excel file and statistical analyses were performed once the file was complete. Two authors (P.D.L. and A.R.) analyzed the data from all centers and then shared the results with the other researchers for review and conclusion-drawing. Table 1 shows the epidemiological and clinical features of the patients included in the study.

### 2.2. Immunohistochemistry

Four-micrometer-thick tissue sections were mounted on poly-L-lysine-coated slides and analyzed using immunohistochemistry (IHC) with a monoclonal anti-BAG3 antibody developed in our laboratories. The anti-BAG3 monoclonal antibody (mAb) was generated by immunizing four mice with a recombinant polypeptide encompassing amino acids 89 to 213 of the human BAG3 protein. Anti-BAG3 antibody titers in the mice sera were monitored using the ELISA method. Subsequently, myeloma cells were fused with splenocytes harvested from the sacrificed mice, followed by screening, selection, and expansion of positive clones producing antibodies specific to the target protein. This standard process successfully yielded the clone utilized in the present study. The IHC protocol included deparaffinization in xylene, rehydration through a graded series of ethanol to water, incubation with 3% hydrogen peroxide for 5 min to inactivate endogenous peroxidases, and enzymatic antigen retrieval in CC1 buffer (Ventana Medical System), pH 8.0, for 36 min at 95 °C. After washing with saline solution (1X PBS), the samples were blocked with 5% fetal bovine serum in 0.1% PBS/BSA and then incubated for 1 h at room temperature with the anti-BAG3 antibody (3 μg/mL) or with murine IgG as isotype control. The reaction was developed using the standard streptavidin-biotin technique, with 3,3′-diaminobenzidine (DAB) as the substrate/chromogen for peroxidase activity. Finally, cell nuclei were counterstained with hematoxylin, and coverslips were mounted using a synthetic mounting medium. BAG3 staining was negative in the non-neoplastic components of tissue samples and was also absent in 24 tested samples from human parotid tissue. In cancer samples tested, immunoreactivity was assessed using a semi-quantitative scoring system: score 0 (complete negativity), score 1 (weak reaction), score 2 (moderate reaction), and score 3 (strong reaction). In cases where there was uncertainty between scores 1 and 2, the case was reviewed collectively by the research team to reach a consensus. These scoring thresholds were applied consistently across all tissue types analyzed.

### 2.3. Statistical Analysis

Statistical correlation between BAG3 expression and patient data were performed using MedCalc^®^ Statistical Software version 20.115 (MedCalc Software Ltd., Ostend, Belgium). Fisher’s exact test was used to test correlations between discrete variables. For survival analysis, Kaplan–Meier estimation with a Log-rank test and Cox proportional hazards models were applied. All tests were two-sided, and *p*-values < 0.05 were considered statistically significant.

### 2.4. Ethical Considerations

Ethical approval was not required by the local ethics committee due to the retrospective and non-interventional design of the study, in accordance with the Italian legislation. The study adhered to the principles outlined by the Helsinki Declaration. All data gathered were entered into an electronic database. Patients who were alive at the time of the enrollment were contacted by phone and informed about the study; none declined participation.

## 3. Results

Following immunohistochemical staining, 104 paraffin-embedded primary tumor samples from therapy-naive patients were evaluated for BAG3 protein expression. The analysis encompassed 30 samples from oral cavity tumors, 8 from oropharynx tumors, and 66 from larynx tumors. Among these specimens, 14 were negative for BAG3 expression, while the remaining samples tested positive. Figure 1 illustrates the immunohistochemical scoring system for BAG3 expression, ranging from negative (no detectable staining) to high positivity (strong and widespread staining). Specifically, it shows representative tumor tissue samples stained with a monoclonal anti-BAG3 antibody, including examples of BAG3-negative samples and positive samples categorized by increasing staining intensity and extent as low positivity (BAG3 score 1), medium positivity (BAG3 score 2), and high positivity (BAG3 score 3).

BAG3 scores did not show significant correlations with sex (*p* = 0.903), smoking status (*p* = 0.808), or HPV infection (*p* = 0.321). When comparing BAG3 immunostaining scores across tumors of different localizations, a relatively even distribution of negative and positive samples at different intensity signal was observed. Specifically, the analysis revealed no significant correlation between BAG3 positivity and any anatomical site of HNSCC (*p* = 0.729) (Figure 2A). The distribution of BAG3 score immunoreactivity was also evaluated across patients with tumor grades ranging from low (G1) to high (G3). No statistically significant differences were observed in the distribution of these scores across the different tumor grades (*p* = 0.540). The same observation was made when analyzing the data distribution across T (*p* = 0.356) and N (*p* = 0.235) parameters (Figure 2C,D). The observed results indicate that BAG3 negative and positive tissues are broadly and diffusely distributed across classifications of HNSCC tumors (Table 2).

Given the potential role of BAG3 in driving molecular phenotypes towards more aggressive and resistant tumors, its prognostic value was investigated. Disease-free survival was correlated with available clinic-pathological patient data and BAG3 immunoreactivity obtained scores. Data collected from 104 patients during the observational period showed a mean recurrence rate of 37.5%. A significant increase in hazard ratio (HR) was observed in our cohort in relation to tumor grade (G2 vs. G1: HR = 3.4; G3 vs. G1: HR = 3.31). Although this parameter is considered prognostically and therapeutically significant in numerous cancer types, its role in HNSCC remains controversial and is not used as a staging criterion [32]. A significant association between the TNM T parameter was found only when correlating T1 with T4 survival data, as well as N0 with N3. Notably, 54.2% of patients with high BAG3 expression experienced recurrence, compared to 36.6% of patients with medium or low BAG3 scores. In contrast, only 14.3% of BAG3-negative patients showed disease recurrence. Patients with the highest BAG3 score had a shorter disease-free survival (median: 23.2 months) compared to BAG3-negative patients (median: 31.3 months) (*p* = 0.014) (Figure 3). Cox proportional hazards analysis further demonstrated that high BAG3 expression (score 3) was associated with more than a threefold increased risk of disease recurrence (univariate analysis: hazard ratio 3.74; 95% confidence interval, 1.30–10.77). Patients with intermediate BAG3 scores (score 1 and score 2) had mean disease-free survival times of 23.8 and 23.2 months, respectively—both shorter than BAG3-negative patients—implying a possible trend which needs further investigation.

## 4. Discussion

HNSCC is notably challenging to treat due to its aggressive nature, complex anatomical location, and high recurrence rates [33,34]. Despite advancements in surgical techniques and adjuvant therapies (RT/ChT), recurrence remains the leading cause of mortality in HNSCC patients [35,36]. Recurrence is a significant challenge, with approximately 50–60% of patients experiencing locoregional recurrence within two years post-treatment, and 20–30% developing distant metastases [37]. Therapeutic options for recurrent or metastatic HNSCC are limited and often result in poor outcomes. Platinum-based chemotherapies and immunotherapies, such as checkpoint inhibitors, offer modest survival benefits [38]. As research progresses towards more effective treatments, tissue biomarkers play a crucial role in patient risk stratification. Incorporating biomarkers into clinical practice can significantly enhance the precision of risk assessment and personalize therapeutic approaches. This strategy aims to reduce recurrence rates and improve patient outcomes [39]. Indeed, recent advancements in understanding prognostic markers and personalized treatment strategies for HNSCC emphasize the integration of molecular, clinical, and histological data. As a first example, the Accurate Prediction Model of HNSCC Overall Survival Score (APMHO), developed using transcriptional data of six key cancer driver genes (CRLF2, HSP90AA1, MAP2K1, PAFAH1B2, MYCL, and SET) alongside clinical variables like age and tumor stage, demonstrated robust predictive performance and highlighted distinct molecular and immune profiles between high- and low-APMHO groups [40]. Another study underscored the prognostic value of tumor budding (TB) as an independent biomarker in both HPV-positive and HPV-negative oropharyngeal cancer. By optimizing TB cutpoints, a superior prognostic accuracy was achieved, enabling refined risk stratification [41]. Another tumor tissue characteristic, as tumor-infiltrating lymphocyte (TIL) counts, assessed via hematoxylin and eosin (H&E)-stained samples, was studied by Torri et al. [42], who found that it was strongly associated with improved survival outcomes (OS, DFS, and DSS), demonstrating their value as a cost-effective biomarker for routine pathology workflows. Finally, histological tumor grade also emerged as a critical predictor of immunotherapy response, with high-grade tumors exhibiting increased sensitivity to immune checkpoint inhibitors, likely due to higher tumor mutational burden [43]. The evidence presented highlights the growing potential of integrating diverse prognostic markers to enhance risk stratification, refine survival predictions, and inform personalized therapeutic strategies for HNSCC. In this study, BAG3 expression was evaluated in 104 HNSCC samples from various tumor sites, and samples with high BAG3 expression were found to have a significantly higher risk of recurrence compared to BAG3-negative samples. BAG3 protein has also been described as a serum biomarker in pancreatic cancer [44], as well as in other illnesses characterized by inflammation, such as psoriasis [45], and fibrosis, such as systemic sclerosis [46,47]. Extracellular BAG3 significantly influences the tumor microenvironment by promoting fibrosis and activating stromal cells, as demonstrated in pancreatic cancer [25,26,27,28]. This fibrotic remodeling supports tumor progression and creates an immunosuppressive environment that hinders effective therapy. Neutralizing BAG3 with monoclonal antibodies has been shown to reduce fibrosis, inhibit tumor growth, and enhance the efficacy of immunotherapy by improving immune cell infiltration and activation. Given the similarities in tumor–stroma interactions, BAG3 may play a comparable role in head and neck squamous cell carcinoma (HNSCC). Therefore, targeting extracellular BAG3 holds potential as a novel therapeutic strategy to improve treatment outcomes in HNSCC.

### Study Limitations

When assessing the importance and generalizability of the data given in this study, numerous limitations must be considered. As a retrospective analysis, the study has inherent methodological limitations. While the sample size appears small for a general oncology cohort, it is very representative considering the rarity of HNSCC, especially oral and oropharyngeal squamous cell carcinoma. Furthermore, like with other potential tissue biomarkers, the findings presented here need to be validated in larger clinical populations. Future research will be required to completely establish the clinical value of these techniques and confirm their role in guiding HNSCC patients’ care.

## 5. Conclusions

This study is inherently exploratory in nature, as it is the first investigation to our knowledge into the possible involvement of the BAG3 protein as a tissue biomarker related to disease recurrence in HNSCC patients. While the findings are promising, they are still preliminary and highlight the need for additional research. Prospective, multicenter studies are required to evaluate BAG3’s prognostic importance and investigate its potential as a molecular biomarker for disease progression monitoring and adjuvant therapy response prediction. Such developments may open the door for more precise and individualized ways to manage HNSCC.

## Figures and Tables

**Figure 1 cancers-17-01843-f001:**
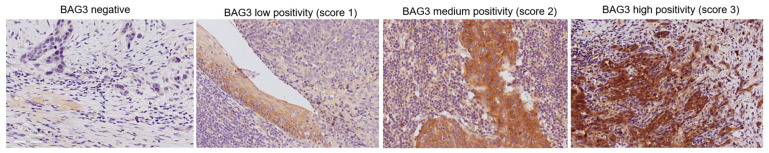
Representative images from different tumor localizations are displayed based on BAG3 immunoreactivity score (images were captured at 20× magnification).

**Figure 2 cancers-17-01843-f002:**
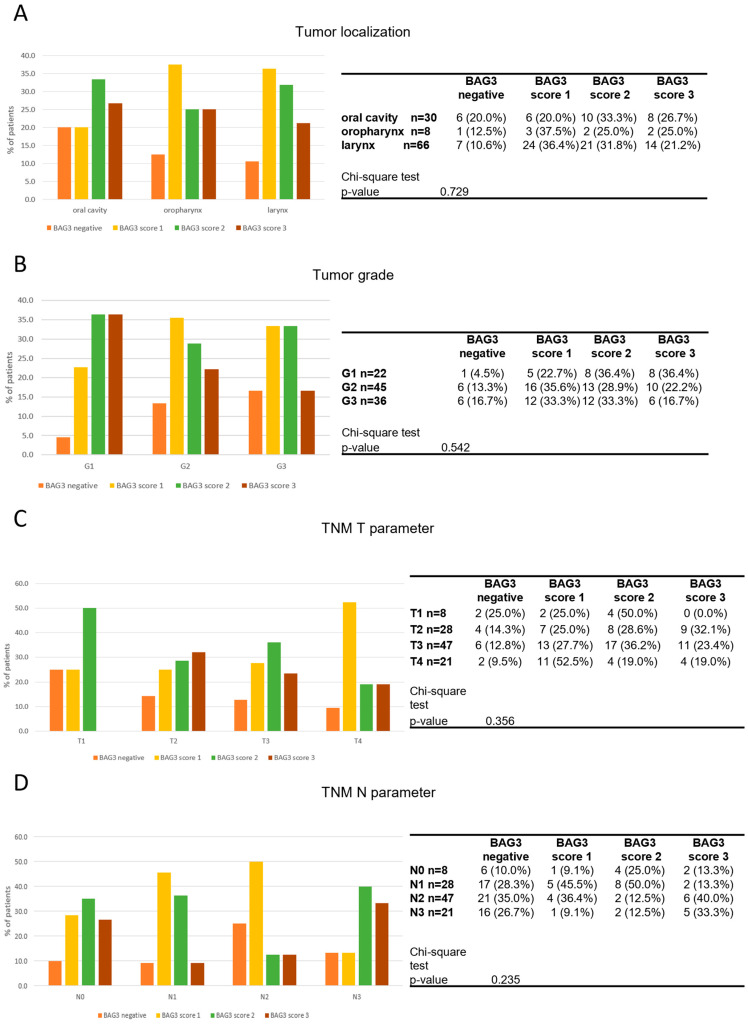
Distribution of BAG3 score obtained in tumor samples across tumor localization (**A**), tumor grade (**B**), TNM T parameter (**C**) and TNM N parameter (**D**). Groups were compared with appropriate contingency tables and by fisher’s exact test. The graphs and tables show the case numbers.

**Figure 3 cancers-17-01843-f003:**
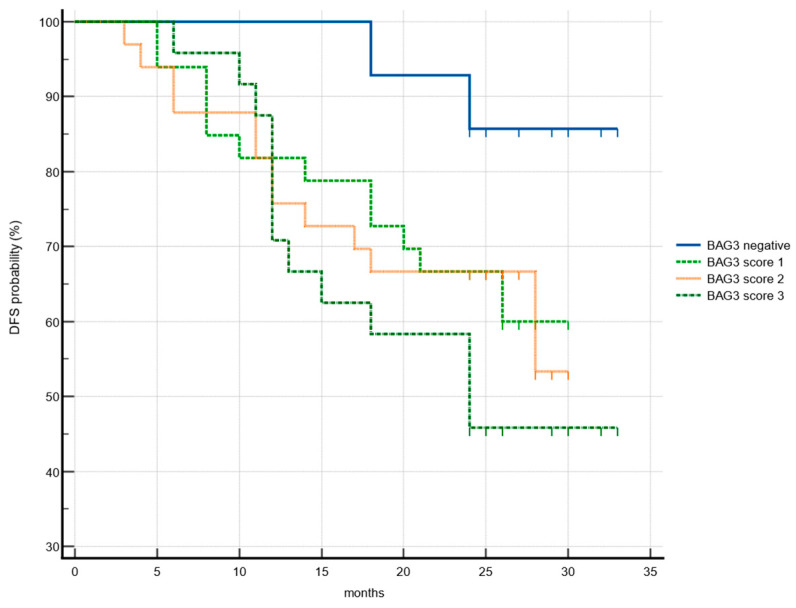
Association of disease-free survival (DFS) with BAG3 immunoreactivity. Kaplan–Meier plots DFS according to the groups of BAG3 negative (blue), BAG3 score 1 (light green), BAG3 score 2 (orange), BAG3 score 3 (dark green).

**Table 1 cancers-17-01843-t001:** Patients characteristics at the time of diagnosis.

Variable	No. Patients	Variable	No. Patients
Total no. of patients = 104			
**Sex**		**TNM** **^1^ T parameter**	
Male	74	T1	8
Female	30	T2	28
**Age (year)**		T3	47
Mean	65 (32–90)	T4	21
**Tobacco use**		**TNM** **^1^ N parameter**	
No	14	N0	60
Yes	63	N1	11
No data	27	N2	16
**Localization**		N3	15
Oral Cavity	30	No data	2
Oropharynx	8	**Grade**	
Larynx	66	G1	22
**HPV infection**		G2	45
Negative	30	G3	36
Positive	8	No data	1
No data	66		

^1^ TNM: postoperative staging, UICC TNM 8th edition.

**Table 2 cancers-17-01843-t002:** Multivariate Cox regression analysis.

Variable	HR	95% CI	*p* Value
Sex, M vs. F	1.43	0.70–2.93	0.320
Tumor grade, G2 vs. G1	3.24	1.34–7.82	<0.001
Tumor grade, G3 vs. G1	3.31	1.30–8.26	<0.001
Local tumor stage, T2 vs. T1	1.8	0.54–5.99	0.550
Local tumor stage, T3 vs. T1	3.92	1.23–12.4	0.160
Local tumor stage, T4 vs. T1	6.39	1.77–23.06	0.024
Node stage, N1 vs. N0	0.81	0.29–2.20	0.770
Node stage, N2 vs. N0	1.44	0.58–3.33	0.400
Node stage, N3 vs. N0	2.45	0.93–6.43	0.019
BAG3 high positive (score 3) vs. BAG3 negative	3.74	1.30–10.77	0.014

## Data Availability

Data are available from the corresponding author upon reasonable request.

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
