# Peer review of "BAG3 Positivity as Prognostic Marker in Head and Neck Squamous Cell Carcinoma"

_cancers, 2025, doi:10.3390/cancers17111843_

Round 1
Reviewer 1 Report
Comments and Suggestions for Authors
The manuscript titled “BAG3 Positivity as Prognostic Marker in Head and Neck Squamous Cell Carcinoma” by Pietro De Luca et al. demonstrated that BAG3 positivity could aid in diagnosing tumor progression and prognosis in head and neck squamous cell carcinoma, with potential implications for therapeutic strategies. Head and neck squamous cell carcinoma is notably challenging to treat due to its aggressive nature, complex anatomical location, and high recurrence rates. Therapeutic options for recurrent or metastatic head and neck squamous cell carcinoma are limited and often result in poor outcomes. Thus, the biomedical rationale for this study is sound and interest. However, there are some comments for improvement.
Comments:
- in the method 2.2 Immunohistochemistry “For the evaluation of BAG3 immunoreactions, the most reproducible threshold for all assessors was established by categorizing samples as either negative or positive. Positive samples were those that exhibited a medium to high intensity of the signal”. The authors should give the standard for negative or positive. No BAG3 expression is negative? In figure 1, BAG3 negative in larynx is different form that in oral cavity or oropharynx.
- In the immunohistochemical data, more clearly images should be provided for figure 1. It is hard to get the details in the current images.
- In this study, P value of disease free survival is 0.054 in figure 3. Is it statistically significant?
Author Response
REVIEWER 1
The manuscript titled “BAG3 Positivity as Prognostic Marker in Head and Neck Squamous Cell Carcinoma” by Pietro De Luca et al. demonstrated that BAG3 positivity could aid in diagnosing tumor progression and prognosis in head and neck squamous cell carcinoma, with potential implications for therapeutic strategies. Head and neck squamous cell carcinoma is notably challenging to treat due to its aggressive nature, complex anatomical location, and high recurrence rates. Therapeutic options for recurrent or metastatic head and neck squamous cell carcinoma are limited and often result in poor outcomes. Thus, the biomedical rationale for this study is sound and interest. However, there are some comments for improvement.
COMMENT 1: in the method 2.2 Immunohistochemistry “For the evaluation of BAG3 immunoreactions, the most reproducible threshold for all assessors was established by categorizing samples as either negative or positive. Positive samples were those that exhibited a medium to high intensity of the signal”. The authors should give the standard for negative or positive. No BAG3 expression is negative? In figure 1, BAG3 negative in larynx is different form that in oral cavity or oropharynx.
RESPONSE 1: We thank the reviewer for highlighting this important point. We have revised the manuscript to provide a more comprehensive explanation of the criteria used to classify samples according to anti-BAG3 immunoreactivity. Specifically, immunoreactivity was evaluated using a semi-quantitative scoring system: score 0 (complete negativity), score 1 (weak reaction), score 2 (moderate reaction), and score 3 (strong reaction). In cases where there was uncertainty between scores 1 and 2, the case was reviewed collectively by the research team to reach a consensus. These scoring thresholds were applied consistently across all tissue types analyzed (see text lines 171-177). Furthermore, we optimized Figure 1 to improve clarity of details. Representative images have been carefully selected and assembled to clearly illustrate the range of immunoreactivity scores observed in the analyzed samples.
COMMENT 2: In the immunohistochemical data, more clearly images should be provided for figure 1. It is hard to get the details in the current images.
RESPONSE 2: As clarified in our previous response, Figure 1 has been revised to include representative images that more clearly and simply illustrate the full range of immunoreactivity.
COMMENT 3: In this study, P value of disease-free survival is 0.054 in figure 3. Is it statistically significant?
RESPONSE 3: We agree with the reviewer’s concern. Although we do not reject the null hypothesis at the conventional 0.050 significance level, the observed p-value of 0.054 is very close to this threshold. By reanalyzing the data and presenting more detailed immunoreactivity scores, we were able to highlight a clearer positive correlation between the proposed marker and tumor recurrence (see new text lines 231-241). Overall, the collected data suggest a trend toward a correlation between BAG3 protein expression in tumor cells and recurrence, indicating that further investigations are needed.
Reviewer 2 Report
Comments and Suggestions for Authors
The study is interesting. Authors investigated BAG3 protein as a biomarker related to HNSCC disease recurrence. Since BAG 3 has been reported as drug resistance marker in various cancers, evaluating its expression especially by IHC in HNSCC is interesting. It is known that HPV positive HNSCCs have better prognosis and treatment outcome than HPV negative, it would be very useful to evaluate BAG3 in HPV +ve Vs. HPV -ve cases of HNSCC. In this study authors showed no significant correlation of BAG3 positivity with HPV infection, smoking, and sex. BAG3 positivity is defined as medium to high intensity. Stratifying BAG3 positivity as low, medium and high and evaluating correlation in subgroups of HNSCC may be a better way to determine correlation of BAG3 positivity especially HPV positive oropharyngeal cancers.
- BAG3 antibody generated in the laboratory. Need to provide some details on the methods how BAG3 antibody is produced in the lab.
- Since BAG3 antibody generated in the lab, is the antibody tested with positive and negative control? Has the isotype control was used? These details are not provided in the methods.
- TCGA data base on BAG3 mRNA expression shows clearly the mRNA expression of BAG3 is significantly lower in HPV+ Ve primary tumors compared to HPV -Ve primary tumors. IHC data from the cohort used in this study showed no correlation.
- Is the HPV correlation was done with including all the subtypes utilized in the study? What would be the correlation if the study includes only Oropharyngeal subtype which has majority of HPV +ve cases.
- BAG3 positivity is defined as medium to high intensity signal. would it be possible to define positivity as low, medium, and high and perform statistical analysis for correlation? This stratification may provide positive correlation.
Author Response
REVIEWER 2
The study is interesting. Authors investigated BAG3 protein as a biomarker related to HNSCC disease recurrence. Since BAG 3 has been reported as drug resistance marker in various cancers, evaluating its expression especially by IHC in HNSCC is interesting. It is known that HPV positive HNSCCs have better prognosis and treatment outcome than HPV negative, it would be very useful to evaluate BAG3 in HPV +ve Vs. HPV -ve cases of HNSCC. In this study authors showed no significant correlation of BAG3 positivity with HPV infection, smoking, and sex. BAG3 positivity is defined as medium to high intensity. Stratifying BAG3 positivity as low, medium and high and evaluating correlation in subgroups of HNSCC may be a better way to determine correlation of BAG3 positivity especially HPV positive oropharyngeal cancers.
COMMENT 1: BAG3 antibody generated in the laboratory. Need to provide some details on the methods how BAG3 antibody is produced in the lab.
RESPONSE 1: We thank the reviewer for requesting more details about the antibody generated. Information regarding the selected BAG3 immunogen and the protocols used to produce the monoclonal antibody are provided in the Materials and Methods section 2.2 (see text lines 155-160).
COMMENT 2: Since BAG3 antibody generated in the lab, is the antibody tested with positive and negative control? Has the isotype control was used? These details are not provided in the methods.
RESPONSE 2: We have now added further details in the Materials and Methods section 2.2 (see text lines 171-172). Specifically, we clarified that the immunohistochemistry procedure was optimized to avoid nonspecific staining, as confirmed using isotype controls (murine IgG antibodies). Additionally, BAG3 staining was negative in the non-neoplastic components of tissue samples and was also absent in 24 tested samples from human parotid tissue.
COMMENT 3: TCGA data base on BAG3 mRNA expression shows clearly the mRNA expression of BAG3 is significantly lower in HPV+ Ve primary tumors compared to HPV -Ve primary tumors. IHC data from the cohort used in this study showed no correlation. Is the HPV correlation was done with including all the subtypes utilized in the study? What would be the correlation if the study includes only Oropharyngeal subtype which has majority of HPV +ve cases.
RESPONSE 3: We analyzed the HPV correlation only in oropharyngeal cancers, as HPV expression in oral and laryngeal carcinomas is rare. The number of HPV-positive oropharyngeal cases in our study was too small for meaningful statistical analysis. Additionally, we have discussed existing literature data on the correlation between BAG3 expression and HPV positivity (see text lines 106-112).
COMMENT 4: BAG3 positivity is defined as medium to high intensity signal. would it be possible to define positivity as low, medium, and high and perform statistical analysis for correlation? This stratification may provide positive correlation.
RESPONSE 4: We sincerely appreciated your suggestion, as it encouraged us to conduct a more thorough analysis of the collected data. Specifically, immunoreactivity was evaluated using a semi-quantitative scoring system: score 0 (complete negativity), score 1 (weak reaction), score 2 (moderate reaction), and score 3 (strong reaction). In cases where the distinction between scores 1 and 2 was ambiguous, the research team collectively reviewed the samples to reach a consensus. These scoring criteria were applied consistently across all tissue types analyzed (see text lines 171- 177). Furthermore, we have included additional correlation data in the Results section (see text lines 231-237), which revealed a clear trend of increasing recurrence rates with higher BAG3 expression in tumor tissues. Notably, the hazard ratio for recurrence in highly positive samples compared to negative tumors was 3.74 (p = 0.014).
Reviewer 3 Report
Comments and Suggestions for Authors
The manuscript “BAG3 Positivity as Prognostic Marker in Head and Neck Squamous Cell Carcinoma” by De Luca et al. aims to investigate BAG3 as a potential biomarker for HNSCC recurrence.
The rationale and aims are clearly stated. The methods used are easily reproducible. The topic of the manuscript is quite original and relevant. In fact, to the best of my knowledge, this is the first study exploring the potential role of BAG3 in HNCs. In addition, the authors have an established track record in the field of BAG3 and its involvement in cancer, which strengthens the credibility of the work.
Minor concerns:
- In general, the references are appropriate and relevant. However, I noticed that 15 out of the 43 references are self-citations. While I understand that self-citations appear pertinent and justified given the authors’ expertise, I would recommend including additional references regarding the role of BAG3 in tumors to further strengthen the background and context of this study.
Overall, although the study is relatively simple in its design and scope, it provides a valuable initial insight into the potential role of BAG3 in HNC recurrence.
Author Response
REVIEWER 3:
The manuscript “BAG3 Positivity as Prognostic Marker in Head and Neck Squamous Cell Carcinoma” by De Luca et al. aims to investigate BAG3 as a potential biomarker for HNSCC recurrence. The rationale and aims are clearly stated. The methods used are easily reproducible. The topic of the manuscript is quite original and relevant. In fact, to the best of my knowledge, this is the first study exploring the potential role of BAG3 in HNCs. In addition, the authors have an established track record in the field of BAG3 and its involvement in cancer, which strengthens the credibility of the work.
Minor concerns:
COMMENT 1: In general, the references are appropriate and relevant. However, I noticed that 15 out of the 43 references are self-citations. While I understand that self-citations appear pertinent and justified given the authors’ expertise, I would recommend including additional references regarding the role of BAG3 in tumors to further strengthen the background and context of this study. Overall, although the study is relatively simple in its design and scope, it provides a valuable initial insight into the potential role of BAG3 in HNC recurrence.
RESPONSE 1: We appreciate the reviewer’s insightful comment and have incorporated the appropriate references (17 and 21) to adequately fulfill the suggestion
Reviewer 4 Report
Comments and Suggestions for Authors
The authors conducted a multi-institutional retrospective study focusing on BAG3 expression by immunohistochemistry in 104 tissue samples from patients with head and neck squamous cell carcinoma (HNSCC), aiming to elucidate the correlation of BAG3 and cancer progression.
The HNSCC prognostic marker study is meaningful for researchers and physicians, and the results are clearly elucidated to the readers.
However, the main flaw of this manuscript is that the sample size is small for a retrospective study, especially since 3 types of HNSCC were included.
Another concern is how reliable we can trust the BAG3 positivity as a prognostic marker. There might be sufficient public RNA sequencing data in HNSCC for the authors to validate the assumption.
Lastly, I know that the authors only detected positive/negative BAG3 expression in the slides. How about testing the molecular expression levels of BAG3 and correlating the BAG3 expression with survival?
Other comments are below.
- Table 1. When correlating BAG3 positivity with variables in Table 1, how did the authors deal with patients with no tobacco use/no HPV infection data?
- If the authors had the patients' recurrence information, how would the BAG3 positivity correlate with the recurrence rate?
Author Response
REVIEWER 4
The authors conducted a multi-institutional retrospective study focusing on BAG3 expression by immunohistochemistry in 104 tissue samples from patients with head and neck squamous cell carcinoma (HNSCC), aiming to elucidate the correlation of BAG3 and cancer progression. The HNSCC prognostic marker study is meaningful for researchers and physicians, and the results are clearly elucidated to the readers.
COMMENT 1: However, the main flaw of this manuscript is that the sample size is small for a retrospective study, especially since 3 types of HNSCC were included.
RESPONSE 1: The study was conducted on 104 tumor samples of the same cell type, despite originating from three different anatomical sites. We believe that the total number of samples is sufficiently representative for a preliminary study in HNSCC.
COMMENT 2: Another concern is how reliable we can trust the BAG3 positivity as a prognostic marker. There might be sufficient public RNA sequencing data in HNSCC for the authors to validate the assumption.
RESPONSE 2: We appreciate this comment and have used it as an opportunity to more thoroughly discuss previous data published by our group regarding the correlation of BAG3 gene expression across various fibrotic tumors. The relevant reference (see text lines 113- 118) has been detailed in the Introduction.This analysis, based on the GEPIA database, provided BAG3 mRNA expression and survival data across multiple cancer types, including head and neck squamous cell carcinoma (HNSCC), allowing comparison of BAG3 levels between tumor and normal tissues and revealing that higher BAG3 expression correlates with poorer clinical outcomes. These findings further motivated us to investigate whether a tool such as immunohistochemistry (IHC) for BAG3 could be useful as a prognostic factor in patients.
COMMENT 3: Lastly, I know that the authors only detected positive/negative BAG3 expression in the slides. How about testing the molecular expression levels of BAG3 and correlating the BAG3 expression with survival?
RESPONSE 3: We appreciate the reviewer’s suggestion, but the present study was aimed mainly at verifying BAG3 protein expression by immunohistochemistry. On the other hand, the correlation between BAG3 mRNA levels and overall survival was already commented in our previous paper (De Marco, M. et al., 2021) and further commented in the introduction.
COMMENT 4:
Table 1. When correlating BAG3 positivity with variables in Table 1, how did the authors deal with patients with no tobacco use/no HPV infection data?
RESPONSE 4: Thank you for highlighting this issue. Unfortunately, data regarding the mentioned risk factors are available for only a subset of patients, and there are only three patients in the entire dataset who are negative for both smoking and HPV. Therefore, we are unable to identify any significant differences in behavior within the examined cohort.
COMMENT 5: If the authors had the patients' recurrence information, how would the BAG3 positivity correlate with the recurrence rate?
RESPONSE 5: We have now added detailed information on the recurrence rate based on BAG3 immunoreactivity score (see text lines 231- 234).
Round 2
Reviewer 1 Report
Comments and Suggestions for Authors
No more comments